# Modelling sparsity, heterogeneity, reciprocity and community structure in temporal interaction data

**Xenia Miscouridou[1], François Caron[1], Yee Whye Teh[1,2]**
[1]Department of Statistics, University of Oxford
[2]DeepMind
{miscouri, caron, y.w.teh}@stats.ox.ac.uk

## Abstract

We propose a novel class of network models for temporal dyadic interaction data. Our objective is to capture important features often observed in social interactions: sparsity, degree heterogeneity, community structure and reciprocity. We use mutually-exciting Hawkes processes to model the interactions between each (directed) pair of individuals. The intensity of each process allows interactions to arise as responses to opposite interactions (reciprocity), or due to shared interests between individuals (community structure). For sparsity and degree heterogeneity, we build the non time dependent part of the intensity function on compound random measures following (Todeschini et al., 2016). We conduct experiments on real-world temporal interaction data and show that the proposed model outperforms competing approaches for link prediction, and leads to interpretable parameters.

## 1 Introduction

There is a growing interest in modelling and understanding temporal dyadic interaction data. Temporal interaction data take the form of time-stamped triples $(t, i, j)$ indicating that an interaction occurred between individuals $i$ and $j$ at time $t$. Interactions may be directed or undirected. Examples of such interaction data include commenting a post on an online social network, exchanging an email, or meeting in a coffee shop. An important challenge is to understand the underlying structure that underpins these interactions. To do so, it is important to develop statistical network models with interpretable parameters, that capture the properties which are observed in real social interaction data.

One important aspect to capture is the *community structure* of the interactions. Individuals are often affiliated to some latent communities (e.g. work, sport, etc.), and their affiliations determine their interactions: they are more likely to interact with individuals sharing the same interests than to individuals affiliated with different communities. An other important aspect is *reciprocity*. Many events are responses to recent events of the opposite direction. For example, if Helen sends an email to Mary, then Mary is more likely to send an email to Helen shortly afterwards. A number of papers have proposed statistical models to capture both community structure and reciprocity in temporal interaction data (Blundell et al., 2012; Dubois et al., 2013; Linderman and Adams, 2014). They use models based on Hawkes processes for capturing reciprocity and stochastic block-models or latent feature models for capturing community structure.

In addition to the above two properties, it is important to capture the global properties of the interaction data. Interaction data are often *sparse*: only a small fraction of the pairs of nodes actually interact. Additionally, they typically exhibit high degree (number of interactions per node) *heterogeneity*: some individuals have a large number of interactions, whereas most individuals have very few, therefore resulting in empirical degree distributions being heavy-tailed. As shown by Karrer and Newman (2011), Gopalan et al. (2013) and Todeschini et al. (2016), failing to account explicitly for

degree heterogeneity in the model can have devastating consequences on the estimation of the latent structure.

Recently, two classes of statistical models, based on random measures, have been proposed to capture sparsity and power-law degree distribution in network data. The first one is the class of models based on exchangeable random measures (Caron and Fox, 2017; Veitch and Roy, 2015; Herlau et al., 2016; Borgs et al., 2018; Todeschini et al., 2016; Palla et al., 2016; Janson, 2017a). The second one is the class of edge-exchangeable models (Crane and Dempsey, 2015; 2018; Cai et al., 2016; Williamson, 2016; Janson, 2017b; Ng and Silva, 2017). Both classes of models can handle both sparse and dense networks and, although the two constructions are different, connections have been highlighted between the two approaches (Cai et al., 2016; Janson, 2017b).

The objective of this paper is to propose a class of statistical models for temporal dyadic interaction data that can capture all the desired properties mentioned above, which are often found in real world interactions. These are *sparsity*, *degree heterogeneity*, *community structure* and *reciprocity*. Combining all the properties in a single model is non trivial and there is no such construction to our knowledge. The proposed model generalises existing reciprocating relationships models (Blundell et al., 2012) to the sparse and power-law regime. Our model can also be seen as a natural extension of the classes of models based on exchangeable random measures and edge-exchangeable models and it shares properties of both families. The approach is shown to outperform alternative models for link prediction on a variety of temporal network datasets.

The construction is based on Hawkes processes and the (static) model of Todeschini et al. (2016) for sparse and modular graphs with overlapping community structure. In Section 2, we present Hawkes processes and compound completely random measures which form the basis of our model's construction. The statistical model for temporal dyadic data is presented in Section 3 and its properties derived in Section 4. The inference algorithm is described in Section 5. Section 6 presents experiments on four real-world temporal interaction datasets.

## 2 Background material

### 2.1 Hawkes processes

Let $(t_k)_{k\geq 1}$ be a sequence of event times with $t_k \geq 0$, and let $\mathcal{H}_t = (t_k|t_k \leq t)$ the subset of event times between time 0 and time $t$. Let $N(t) = \sum_{k\geq 1} 1_{t_k \leq t}$ denote the number of events between time 0 and time $t$, where $1_A = 1$ if $A$ is true, and 0 otherwise. Assume that $N(t)$ is a counting process with conditional intensity function $\lambda(t)$, that is for any $t \geq 0$ and any infinitesimal interval $dt$

$$\Pr(N(t + dt) - N(t) = 1|\mathcal{H}_t) = \lambda(t)dt. \tag{1}$$

Consider another counting process $\tilde{N}(t)$ with the corresponding $(\tilde{t}_k)_{k\geq 1}, \tilde{\mathcal{H}}_t, \tilde{\lambda}(t)$. Then, $N(t), \tilde{N}(t)$ are mutually-exciting Hawkes processes (Hawkes, 1971) if the conditional intensity functions $\lambda(t)$ and $\tilde{\lambda}(t)$ take the form

$$\lambda(t) = \mu + \int_0^t g_\phi(t - u) \, d\tilde{N}(u) \qquad \tilde{\lambda}(t) = \tilde{\mu} + \int_0^t g_{\tilde{\phi}}(t - u) \, dN(u)$$

where $\mu = \lambda(0) > 0, \tilde{\mu} = \tilde{\lambda}(0) > 0$ are the base intensities and $g_\phi, g_{\tilde{\phi}}$ non-negative kernels parameterised by $\phi$ and $\tilde{\phi}$. This defines a pair of processes in which the current rate of events of each process depends on the occurrence of past events of the opposite process.

Assume that $\mu = \tilde{\mu}$, $\phi = \tilde{\phi}$ and $g_\phi(t) \geq 0$ for $t > 0$, $g_\phi(t) = 0$ for $t < 0$. If $g_\phi$ admits a form of fast decay then this results in strong local effects. However, if it prescribes a peak away from the origin then longer term effects are likely to occur. We consider here an exponential kernel

$$g_\phi(t - u) = \eta e^{-\delta(t-u)}, t > u \tag{2}$$

where $\phi = (\eta, \delta)$. $\eta \geq 0$ determines the sizes of the self-excited jumps and $\delta > 0$ is the constant rate of exponential decay. The stationarity condition for the processes is $\eta < \delta$. Figure 1 gives an illustration of two mutually-exciting Hawkes processes with exponential kernel and their conditional intensities.

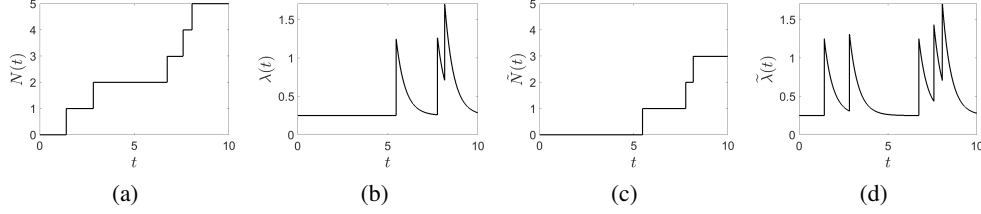

Figure 1: Illustration of mutually-exciting Hawkes processes with exponential kernel with parameters $\mu = 0.25$, $\eta = 1$ and $\delta = 2$. (a) Counting process $N(t)$ and (b) its conditional intensity $\lambda(t)$. (c) Counting process $\tilde{N}(t)$ and its conditional intensity $\tilde{\lambda}(t)$.

## 2.2 Compound completely random measures

A homogeneous completely random measure (CRM) (Kingman, 1967; 1993) on $\mathbb{R}_+$ without fixed atoms nor deterministic component takes the form

$$W = \sum_{i \geq 1} w_i \delta_{\theta_i} \tag{3}$$

where $(w_i, \theta_i)_{i \geq 1}$ are the points of a Poisson process on $(0, \infty) \times \mathbb{R}_+$ with mean measure $\rho(dw)H(d\theta)$ where $\rho$ is a Lévy measure, $H$ is a locally bounded measure and $\delta_x$ is the dirac delta mass at $x$. The homogeneous CRM is completely characterized by $\rho$ and $H$, and we write $W \sim \mathrm{CRM}(\rho, H)$, or simply $W \sim \mathrm{CRM}(\rho)$ when $H$ is taken to be the Lebesgue measure. Griffin and Leisen (2017) proposed a multivariate generalisation of CRMs, called compound CRM (CCRM). A compound CRM $(W_1, \ldots, W_p)$ with independent scores is defined as

$$W_k = \sum_{i \geq 1} w_{ik} \delta_{\theta_i}, \quad k = 1, \ldots, p \tag{4}$$

where $w_{ik} = \beta_{ik} w_{i0}$ and the scores $\beta_{ik} \geq 0$ are independently distributed from some probability distribution $F_k$ and $W_0 = \sum_{i \geq 1} w_{i0} \delta_{\theta_i}$ is a CRM with mean measure $\rho_0(dw_0)H_0(d\theta)$. In the rest of this paper, we assume that $\bar{F}_k$ is a gamma distribution with parameters $(a_k, b_k)$, $H_0(d\theta) = d\theta$ is the Lebesgue measure and $\rho_0$ is the Lévy measure of a generalized gamma process

$$\rho_0(dw) = \frac{1}{\Gamma(1 - \sigma)} w^{-1-\sigma} e^{-\tau w} dw \tag{5}$$

where $\sigma \in (-\infty, 1)$ and $\tau > 0$.

## 3 Statistical model for temporal interaction data

Consider temporal interaction data of the form $\mathcal{D} = (t_k, i_k, j_k)_{k \geq 1}$ where $(t_k, i_k, j_k) \in \mathbb{R}_+ \times \mathbb{N}_*^2$ represents a directed interaction at time $t_k$ from node/individual $i_k$ to node/individual $j_k$. For example, the data may correspond to the exchange of messages between students on an online social network.

We use a point process $(t_k, U_k, V_k)_{k \geq 1}$ on $\mathbb{R}_+^3$, and consider that each node $i$ is assigned some continuous label $\theta_i \geq 0$. the labels are only used for the model construction, similarly to (Caron and Fox, 2017; Todeschini et al., 2016), and are not observed nor inferred from the data. A point at location $(t_k, U_k, V_k)$ indicates that there is a directed interaction between the nodes $U_k$ and $V_k$ at time $t_k$. See Figure 2 for an illustration.

For a pair of nodes $i$ and $j$, with labels $\theta_i$ and $\theta_j$, let $N_{ij}(t)$ be the counting process

$$N_{ij}(t) = \sum_{k|(U_k, V_k) = (\theta_i, \theta_j)} 1_{t_k \leq t} \tag{6}$$

for the number of interactions between $i$ and $j$ in the time interval $[0, t]$. For each pair $i, j$, the counting processes $N_{ij}, N_{ji}$ are mutually-exciting Hawkes processes with conditional intensities

$$\lambda_{ij}(t) = \mu_{ij} + \int_0^t g_\phi(t - u)\, dN_{ji}(u), \qquad \lambda_{ji}(t) = \mu_{ji} + \int_0^t g_\phi(t - u)\, dN_{ij}(u) \tag{7}$$

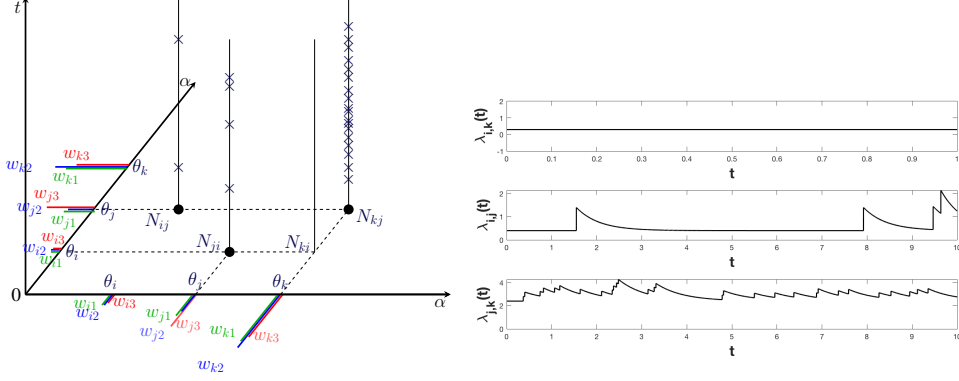

Figure 2: (Left) Representation of the temporal dyadic interaction data as a point process on $\mathbb{R}_+^3$. A point at location $(\tau, \theta_i, \theta_j)$ indicates a directed interaction from node $i$ to node $j$ at time $\tau$, where $\theta_i > 0$ and $\theta_j > 0$ are labels of nodes $i$ and $j$. Interactions between nodes $i$ and $j$, arise from a Hawkes process $N_{ij}$ with conditional intensity $\lambda_{ij}$ given by Equation (7). Processes $N_{ik}$ and $N_{jk}$ are not shown for readability. (Right) Conditional intensities of processes $N_{ik}(t)$, $N_{ij}(t)$ and $N_{jk}(t)$.

where $g_\phi$ is the exponential kernel defined in Equation (2). Interactions from individual $i$ to individual $j$ may arise as a response to past interactions from $j$ to $i$ through the kernel $g_\phi$, or via the base intensity $\mu_{ij}$. We also model assortativity so that individuals with similar interests are more likely to interact than individuals with different interests. For this, assume that each node $i$ has a set of positive latent parameters $(w_{i1}, \ldots, w_{ip}) \in \mathbb{R}_+^p$, where $w_{ik}$ is the level of its affiliation to each latent community $k = 1, \ldots, p$. The number of communities $p$ is assumed known. We model the base rate

$$\mu_{ij} = \mu_{ji} = \sum_{k=1}^p w_{ik} w_{jk}. \tag{8}$$

Two nodes with high levels of affiliation to the same communities will be more likely to interact than nodes with affiliation to different communities, favouring assortativity.

In order to capture sparsity and power-law properties and as in Todeschini et al. (2016), the set of affiliation parameters $(w_{i1}, \ldots, w_{ip})$ and node labels $\theta_i$ is modelled via a compound CRM with gamma scores, that is $W_0 = \sum_{i=1}^\infty w_{i0} \delta_{\theta_i} \sim \mathrm{CRM}(\rho_0)$ where the Lévy measure $\rho_0$ is defined by Equation (5), and for each node $i \geq 1$ and community $k = 1, \ldots, p$

$$w_{ik} = w_{i0} \beta_{ik}, \text{ where } \beta_{ik} \overset{\mathrm{ind}}{\sim} \mathrm{Gamma}(a_k, b_k). \tag{9}$$

The parameter $w_{i0} \geq 0$ is a degree correction for node $i$ and can be interpreted as measuring the overall popularity/sociability of a given node $i$ irrespective of its level of affiliation to the different communities. An individual $i$ with a high sociability parameter $w_{i0}$ will be more likely to have interactions overall than individuals with low sociability parameters. The scores $\beta_{ik}$ tune the level of affiliation of individual $i$ to the community $k$. The model is defined on $\mathbb{R}_+^3$. We assume that we observe interactions over a subset $[0, T] \times [0, \alpha]^2 \subseteq \mathbb{R}_+^3$ where $\alpha$ and $T$ tune both the number of nodes and number of interactions. The whole model is illustrated in Figure 2.

The model admits the following set of hyperparameters, with the following interpretation:
• The hyperparameters $\phi = (\eta, \delta)$ where $\eta \geq 0$ and $\delta \geq 0$ of the kernel $g_\phi$ tune the *reciprocity*.
• The hyperparameters $(a_k, b_k)$ tune the *community structure* of the interactions. $a_k/b_k = \mathbb{E}[\beta_{ik}]$ tunes the size of community $k$ while $a_k/b_k^2 = \mathrm{var}(\beta_{ik})$ tunes the variability of the level of affiliation to this community; larger values imply more separated communities.
• The hyperparameter $\sigma$ tunes the *sparsity* and the *degree heterogeneity*: larger values imply higher sparsity and heterogeneity. It also tunes the slope of the degree distribution. Parameter $\tau$ tunes the exponential cut-off in the degree distribution. This is illustrated in Figure 3.
• Finally, the hyperparameters $\alpha$ and $T$ tune the overall number of interactions and nodes.

We follow (Rasmussen, 2013) and use vague Exponential(0.01) priors on $\eta$ and $\delta$. Following Todeschini et al. (2016) we set vague Gamma(0.01, 0.01) priors on $\alpha$, $1 - \sigma$, $\tau$, $a_k$ and $b_k$. The right limit for time window, $T$ is considered known.

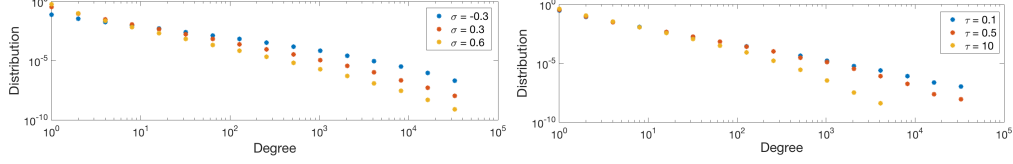

Figure 3: Degree distribution for a Hawkes graph with different values of $\sigma$ (left) and $\tau$ (right). The degree of a node $i$ is defined as the number of nodes with whom it has at least one interaction. The value $\sigma$ tunes the slope of the degree distribution, larger values corresponding to a higher slope. The parameter $\tau$ tunes the exponential cut-off in the degree distribution.

## 4   Properties

### 4.1   Connection to sparse vertex-exchangeable and edge-exchangeable models

The model is a natural extension of sparse vertex-exchangeable and edge-exchangeable graph models. Let $z_{ij}(t) = 1_{N_{ij}(t)+N_{ji}(t)>0}$ be a binary variable indicating if there is at least one interaction in $[0,t]$ between nodes $i$ and $j$ in either direction. We assume

$$\Pr(z_{ij}(t) = 1 | (w_{ik}, w_{jk})_{k=1,\ldots,p}) = 1 - e^{-2t\sum_{k=1}^{p} w_{ik}w_{jk}}$$

which corresponds to the probability of a connection in the static simple graph model proposed by Todeschini et al. (2016). Additionally, for fixed $\alpha > 0$ and $\eta = 0$ (no reciprocal relationships), the model corresponds to a rank-$p$ extension of the rank-1 Poissonized version of edge-exchangeable models considered by Cai et al. (2016) and Janson (2017a). The sparsity properties of our model follow from the sparsity properties of these two classes of models.

### 4.2   Sparsity

The size of the dataset is tuned by both $\alpha$ and $T$. Given these quantities, both the number of interactions and the number of nodes with at least one interaction are random variables. We now study the behaviour of these quantities, showing that the model exhibits sparsity. Let $I_{\alpha,T}, E_{\alpha,T}, V_{\alpha,T}$ be the overall number of interactions between nodes with label $\theta_i \leq \alpha$ until time $T$, the total number of pairs of nodes with label $\theta_i \leq \alpha$ who had at least one interaction before time $T$, and the number of nodes with label $\theta_i \leq \alpha$ who had at least one interaction before time $T$ respectively.

$$V_{\alpha,T} = \sum_i 1_{\sum_{j\neq i} N_{ij}(T)1_{\theta_j \leq \alpha} > 0} 1_{\theta_i \leq \alpha} \qquad I_{\alpha,T} = \sum_{i \neq j} N_{ij}(T) 1_{\theta_i \leq \alpha} 1_{\theta_j \leq \alpha}$$

$$E_{\alpha,T} = \sum_{i<j} 1_{N_{ij}(T)+N_{ji}(T)>0} 1_{\theta_i \leq \alpha} 1_{\theta_j \leq \alpha}$$

We provide in the supplementary material a theorem for the exact expectations of $I_{\alpha,T}, E_{\alpha,T}\ V_{\alpha,T}$. Now consider the asymptotic behaviour of the expectations of $V_{\alpha,T}, E_{\alpha,T}$ and $I_{\alpha,T}$, as $\alpha$ and $T$ go to infinity.[1] Consider fixed $T > 0$ and $\alpha$ that tends to infinity. Then,

$$\mathbb{E}[V_{\alpha,T}] = \begin{cases} \Theta(\alpha) & \text{if } \sigma < 0 \\ \omega(\alpha) & \text{if } \sigma \geq 0 \end{cases}, \qquad \mathbb{E}[E_{\alpha,T}] = \Theta(\alpha^2), \qquad \mathbb{E}[I_{\alpha,T}] = \Theta(\alpha^2)$$

as $\alpha$ tends to infinity. For $\sigma < 0$, the number of edges and interactions grows quadratically with the number of nodes, and we are in the dense regime. When $\sigma \geq 0$, the number of edges and interaction grows subquadratically, and we are in the sparse regime. Higher values of $\sigma$ lead to higher sparsity. For fixed $\alpha$,

$$\mathbb{E}[V_{\alpha,T}] = \begin{cases} \Theta(1) & \text{if } \sigma < 0 \\ \Theta(\log T) & \text{if } \sigma = 0 \\ \Theta(T^\sigma) & \text{if } \sigma > 0 \end{cases}, \qquad \mathbb{E}[E_{\alpha,T}] = \begin{cases} \Theta(1) & \text{if } \sigma < 0 \\ O(\log T) & \text{if } \sigma = 0 \\ O(T^{3\sigma/2}) & \text{if } \sigma \in (0, 1/2] \\ O(T^{(1+\sigma)/2}) & \text{if } \sigma \in (1/2, 1) \end{cases}$$

and $\mathbb{E}[I_{\alpha,T}] = \Theta(T)$ as $T$ tends to infinity. Sparsity in $T$ arises when $\sigma \geq 0$ for the number of edges and when $\sigma > 1/2$ for the number of interactions. The derivation of the asymptotic behaviour of expectations of $V_{\alpha,T}$, $E_{\alpha,T}$ and $I_{\alpha,T}$ follows the lines of the proofs of Theorems 3 and 5.3 in (Todeschini et al., 2016) ($\alpha \to \infty$) and Lemma D.6 in the supplementary material of (Cai et al., 2016) ($T \to \infty$), and is omitted here.

## 5    Approximate Posterior Inference

Assume a set of observed interactions $\mathcal{D} = (t_k, i_k, j_k)_{k \geq 1}$ between $V$ individuals over a period of time $T$. The objective is to approximate the posterior distribution $\pi(\phi, \xi | \mathcal{D})$ where $\phi$ are the kernel parameters and $\xi = ((w_{ik})_{i=1,\dots,V,k=1,\dots,p}, (a_k, b_k)_{k=1,\dots,p}, \alpha, \sigma, \tau)$, the parameters and hyperparameters of the compound CRM. One possible approach is to follow a similar approach to that taken in (Rasmussen, 2013); derive a Gibbs sampler using a data augmentation scheme which associates a latent variable to each interaction. However, such an algorithm is unlikely to scale well with the number of interactions. Additionally, we can make use of existing code for posterior inference with Hawkes processes and graphs based on compound CRMs, and therefore propose a two-step approximate inference procedure, motivated by modular Bayesian inference (Jacob et al., 2017).

Let $\mathcal{Z} = (z_{ij}(T))_{1 \leq i,j \leq V}$ be the adjacency matrix defined by $z_{ij}(T) = 1$ if there is at least one interaction between $i$ and $j$ in the interval $[0, T]$, and 0 otherwise. We have

$$\pi(\phi, \xi | \mathcal{D}) = \pi(\phi, \xi | \mathcal{D}, \mathcal{Z}) = \pi(\xi | \mathcal{D}, \mathcal{Z})\pi(\phi | \xi, \mathcal{D}).$$

The idea of the two-step procedure is to (i) Approximate $\pi(\xi | \mathcal{D}, \mathcal{Z})$ by $\pi(\xi | \mathcal{Z})$ and obtain a Bayesian point estimate $\widehat{\xi}$ then (ii) Approximate $\pi(\phi | \xi, \mathcal{D})$ by $\pi(\phi | \widehat{\xi}, \mathcal{D})$.

The full posterior is thus approximated by $\widetilde{\pi}(\phi, \xi) = \pi(\xi | \mathcal{Z})\pi(\phi | \widehat{\xi}, \mathcal{D})$. As mentioned in Section 4.1, the statistical model for the binary adjacency matrix $\mathcal{Z}$ is the same as in (Todeschini et al., 2016). We use the MCMC scheme of (Todeschini et al., 2016) and the accompanying software package SNetOC[2] to perform inference. The MCMC sampler is a Gibbs sampler which uses a Metropolis-Hastings (MH) step to update the hyperparameters and a Hamiltonian Monte Carlo (HMC) step for the parameters. From the posterior samples we compute a point estimate $(\widehat{w}_{i1}, \dots, \widehat{w}_{ip})$ of the weight vector for each node. We follow the approach of Todeschini et al. (2016) and compute a minimum Bayes risk point estimate using a permutation-invariant cost function. Given these point estimates we obtain estimates of the base intensities $\widehat{\mu}_{ij}$. Posterior inference on the parameters $\phi$ of the Hawkes kernel is then performed using Metropolis-Hastings, as in (Rasmussen, 2013). Details of the two-stage inference procedure are given in the supplementary material.

**Empirical investigation of posterior consistency.** To validate the two-step approximation to the posterior distribution, we study empirically the convergence of our approximate inference scheme using synthetic data. Experiments suggest that the posterior concentrates around the true parameter value. More details are given in the supplementary material.

## 6    Experiments

We perform experiments on four temporal interaction datasets from the Stanford Large Network Dataset Collection[3] (Leskovec and Krevl, 2014):
• The EMAIL dataset consists of emails sent within a large European research institution over 803 days. There are 986 nodes, 24929 edges and 332334 interactions. A separate interaction is created for every recipient of an email.
• The COLLEGE dataset consists of private messages sent over a period of 193 days on an online social network (Facebook-like platform) at the University of California, Irvine. There are 1899 nodes, 20296 edges and 59835 interactions. An interaction $(t, i, j)$ corresponds to a user $i$ sending a private message to another user $j$ at time $t$.
• The MATH overflow dataset is a temporal network of interactions on the stack exchange website Math Overflow over 2350 days. There are 24818 nodes, 239978 edges and 506550 interactions. An

interaction $(t, i, j)$ means that a user $i$ answered another user's $j$ question at time $t$, or commented on another user's $j$ question/response.

• The UBUNTU dataset is a temporal network of interactions on the stack exchange website Ask Ubuntu over 2613 days. There are 159316 nodes, 596933 edges and 964437 interactions. An interaction $(t, i, j)$ means that a user $i$ answered another user's $j$ question at time $t$, or commented on another user's $j$ question/response.

**Comparison on link prediction.** We compare our model (Hawkes-CCRM) to five other benchmark methods: (i) our model, without the Hawkes component (obtained by setting $\eta = 0$), (ii) the Hawkes-IRM model of Blundell et al. (2012) which uses an infinite relational model (IRM) to capture the community structure together with a Hawkes process to capture reciprocity, (iii) the same model, called Poisson-IRM, without the Hawkes component, (iv) a simple Hawkes model where the conditional intensity given by Equation (7) is assumed to be same for each pair of individuals, with unknown parameters $\delta$ and $\eta$, (v) a simple Poisson process model, which assumes that interactions between two individuals arise at an unknown constant rate. Each of these competing models capture a subset of the features we aim to capture in the data: sparsity/heterogeneity, community structure and reciprocity, as summarized in Table 1. The models are given in the supplementary material. The only model to account for all the features is the proposed Hawkes-CCRM model.

Table 1: (Left) Performance in link prediction. (Right) Properties captured by the different models.

| | email | college | math | ubuntu | sparsity/ heterogeneity | community structure | reciprocity |
|---|---|---|---|---|---|---|---|
| Hawkes-CCRM | **10.95** | **1.88** | **20.07** | **29.1** | ✓ | ✓ | ✓ |
| CCRM | 12.08 | 2.90 | 89.0 | 36.5 | ✓ | ✓ | |
| Hawkes-IRM | 14.2 | 3.56 | 96.9 | 59.5 | | ✓ | ✓ |
| Poisson-IRM | 31.7 | 15.7 | 204.7 | 79.3 | | ✓ | |
| Hawkes | 154.8 | 153.29 | 220.10 | 191.39 | | | ✓ |
| Poisson | $\sim 10^3$ | $\sim 10^4$ | $\sim 10^4$ | $\sim 10^4$ | | | |

We perform posterior inference using a Markov chain Monte Carlo algorithm. For our Hawkes-CCRM model, we follow the two-step procedure described in Section 5. For each dataset, there is some background information in order to guide the choice of the number $p$ of communities. The number of communities $p$ is set to $p = 4$ for the EMAIL dataset, as there are 4 departments at the institution, $p = 2$ for the COLLEGE dataset corresponding to the two genders, and $p = 3$ for the MATH and UBUNTU datasets, corresponding to the three different types of possible interactions. We follow Todeschini et al. (2016) regarding the choice of the MCMC tuning parameters and initialise the MCMC algorithm with the estimates obtained by running a MCMC algorithm with $p = 1$ feature with fewer iterations. For all experiments we run 2 chains in parallel for each stage of the inference. We use 100000 iterations for the first stage and 10000 for the second one. For the Hawkes-IRM model, we also use a similar two-step procedure, which first obtains a point estimate of the parameters and hyperparameters of the IRM, then estimates the parameters of the Hawkes process given this point estimate. This allows us to scale this approach to the large datasets considered. We use the same number of MCMC samples as for our model for each step.

We compare the different algorithms on link prediction. For each dataset, we make a train-test split in time so that the training datasets contains $85\%$ of the total temporal interactions. We use the training data for parameter learning and then use the estimated parameters to perform link prediction on the held out test data. We report the root mean square error between the predicted and true number of interactions for each directed pair in the test set . The results are reported in Table 1. On all the datasets, the proposed Hawkes-CCRM approach outperforms other methods. Interestingly, the addition of the Hawkes component brings improvement for both the IRM-based model and the CCRM-based model.

**Community structure, degree distribution and sparsity.** Our model also estimates the latent structure in the data through the weights $w_{ik}$, representing the level of affiliation of a node $i$ to a community $k$. For each dataset, we order the nodes by their highest estimated feature weight, obtaining a clustering of the nodes. We represent the ordered matrix $(z_{ij}(T))$ of binary interactions in Figure 4 (a)-(d). This shows that the method can uncover the latent community structure in the different datasets. Within each community, nodes still exhibit degree heterogeneity as shown in Figure 4 (e)-(h). where the nodes are sorted within each block according to their estimated sociability $\widehat{w}_{i0}$. The ability of the approach to uncover latent structure was illustrated by Todeschini et al. (2016), who demonstrate that models which do not account for degree heterogeneity, cannot capture latent community estimation

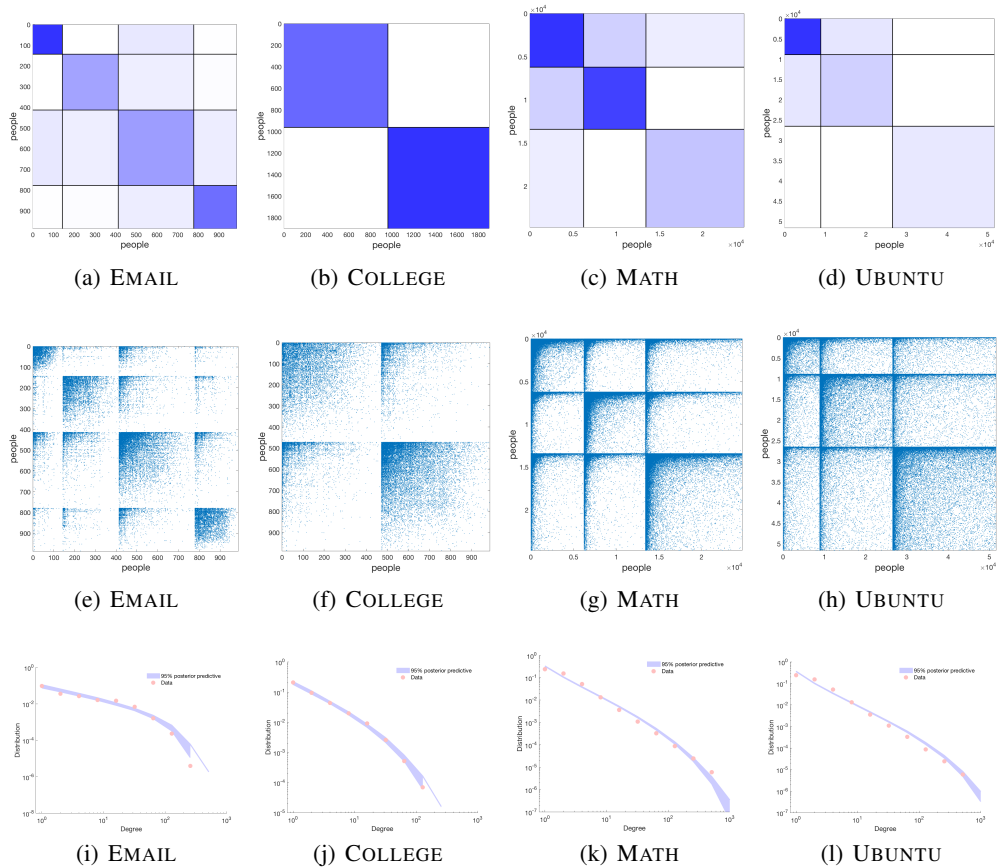

Figure 4: Top: Sorted adjacency matrix for each dataset. The vertices are classified to one of the communities based to their highest affiliation. Darker color correspond to more interactions. Middle: Sorted adjacency matrix. The nodes are grouped according to their highest affiliation and then sorted according to their estimated sociability parameter $\widehat{w}_{i0}$. Bottom: Empirical degree distribution (red) and posterior predictive distribution (blue).

but they rather cluster the nodes based on their degree. We also look at the posterior predictive degree distribution based on the estimated hyperparameters, and compare it to the empirical degree distribution in the test set. The results are reported in Figure 4 (i)-(l) showing a reasonable fit to the degree distribution. Finally we report the $95\%$ posterior credible intervals (PCI) for the sparsity parameter $\sigma$ for all datasets. Each PCI is $(-0.69, -0.50), (-0.35, -0.20), (0.15, 0.18), (0.51, 0.57)$ respectively. The range of $\sigma$ is $(-\infty, 1)$. EMAIL and COLLEGE give negative values corresponding to denser networks whereas MATH and UBUNTU datasets are sparser.

# 7 Conclusion

We have presented a novel statistical model for temporal interaction data which captures multiple important features observed in such datasets, and shown that our approach outperforms competing models in link prediction. The model could be extended in several directions. One could consider asymmetry in the base intensities $\mu_{ij} \neq \mu_{ji}$ and/or a bilinear form as in (Zhou, 2015). Another important extension would be the estimation of the number of latent communities/features $p$.

**Acknowledgments.** The authors thank the reviewers and area chair for their constructive comments. XM, FC and YWT acknowledge funding from the ERC under the European Union's 7th Framework programme (FP7/2007-2013) ERC grant agreement no. 617071. FC acknowledges support from EPSRC under grant EP/P026753/1 and from the Alan Turing Institute under EPSRC grant EP/N510129/1. XM acknowledges support from the A. G. Leventis Foundation.

## Footnotes

[1] We use the following asymptotic notations. $X_\alpha = O(Y_\alpha)$ if $\lim X_\alpha/Y_\alpha < \infty$, $X_\alpha = \omega(Y_\alpha)$ if $\lim Y_\alpha/X_\alpha = 0$ and $X_\alpha = \Theta(Y_\alpha)$ if both $X_\alpha = O(Y_\alpha)$ and $Y_\alpha = O(X_\alpha)$.

[2]https://github.com/misxenia/SNetOC

[3]https://snap.stanford.edu/data/

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
