[Supplementary Material]

# Modelling sparsity, heterogeneity, reciprocity and community structure in temporal interaction data

**Xenia Miscouridou[1], François Caron[1], Yee Whye Teh[1,2]**
[1]Department of Statistics, University of Oxford
[2]DeepMind
{miscouri, caron, y.w.teh}@stats.ox.ac.uk

## 1 Background on compound completely random measures

We give the necessary background on compound completely random measures (CCRM). An extensive account of this class of models is given in (Griffin and Leisen, 2017). In this article, we consider a CCRM $(W_1, \ldots, W_p)$ on $\mathbb{R}_+$ characterized, for any $(t_1, \ldots, t_p) \in \mathbb{R}_+^p$ and measurable set $A \subset \mathbb{R}_+$, by

$$\mathbb{E}[e^{-\sum_{k=1}^p t_k W_k(A)}] = \exp(-H_0(A)\psi(t_1, \ldots, t_p))$$

where $H_0$ is the Lebesgue measure and $\psi$ is the multivariate Laplace exponent defined by

$$\psi(t_1, \ldots, t_p) = \int_{\mathbb{R}_+^p} (1 - e^{-\sum_{k=1}^p t_k w_k})\rho(dw_1, \ldots, dw_p). \tag{1}$$

The multivariate Lévy measure $\rho$ takes the form

$$\rho(dw_1, \ldots, dw_p) = \int_0^\infty w_0^{-p} \prod_{k=1}^p F_k \left( \frac{dw_k}{w_0} \right) \rho_0(dw_0) \tag{2}$$

where $F_k$ is the distribution of a Gamma random variable with parameters $a_k$ and $b_k$ and $\rho_0$ is the Lévy measure on $(0, \infty)$ of a generalized gamma process

$$\rho_0(dw_0) = \frac{1}{\Gamma(1-\sigma)} w_0^{-1-\sigma} \exp(-w_0 \tau) dw_0$$

where $\sigma \in (-\infty, 1)$ and $\tau > 0$.

Denote $\boldsymbol{w}_i = (w_{i1}, \ldots, w_{ip})^T$, $\boldsymbol{\beta}_i = (\beta_{i1}, \ldots, \beta_{ip})^T$ and $\rho(d\boldsymbol{w}) = \rho(dw_1, \ldots, dw_p)$. $w_0$ always refers to the scalar weight corresponding to the measure $\rho_0$.

## 2 Expected number of interactions, edges and nodes

Recall that $I_{\alpha,T}$, $E_{\alpha,T}$ and $V_{\alpha,T}$ are respectively the overall number of interactions between nodes with label $\theta_i \leq \alpha$ until time $T$, the total number of pairs of nodes with label $\theta_i \leq \alpha$ who had at least one interaction before time $T$, and the number of nodes with label $\theta_i \leq \alpha$ who had at least one interaction before time $T$ respectively, and are defined as

$$I_{\alpha,T} = \sum_{i \neq j} N_{ij}(T) 1_{\theta_i \leq \alpha} 1_{\theta_j \leq \alpha}$$

$$E_{\alpha,T} = \sum_{i < j} 1_{N_{ij}(T) + N_{ji}(T) > 0} 1_{\theta_i \leq \alpha} 1_{\theta_j \leq \alpha}$$

$$V_{\alpha,T} = \sum_i 1_{\sum_{j \neq i} N_{ij}(T) 1_{\theta_j \leq \alpha} > 0} 1_{\theta_i \leq \alpha}$$

**Theorem 1** *The expected number of interactions $I_{\alpha,T}$, edges $E_{\alpha,T}$ and nodes $V_{\alpha,T}$ are given as follows:*

$$\mathbb{E}[I_{\alpha,T}] = \alpha^2 \boldsymbol{\mu}_w^T \boldsymbol{\mu}_w \left( \frac{\delta}{\delta - \eta} T - \frac{\eta}{(\eta - \delta)^2} \left( 1 - e^{-T(\delta - \eta)} \right) \right)$$

$$\mathbb{E}[E_{\alpha,T}] = \frac{\alpha^2}{2} \int_{\mathbb{R}_+^p} \psi(2Tw_1, \ldots, 2Tw_p) \rho(dw_1, \ldots, dw_p)$$

$$\mathbb{E}[V_{\alpha,T}] = \alpha \int_{\mathbb{R}_+^p} \left( 1 - e^{-\alpha \psi(2Tw_1, \ldots, 2Tw_p)} \right) \rho(dw_1, \ldots, dw_p)$$

*where $\mu_w = \int_{\mathbb{R}_+^p} \boldsymbol{w} \rho(dw_1, \ldots, dw_p)$.*

The proof of Theorem 1 is given below and follows the lines of Theorem 3 in (Todeschini et al., 2016).

**Mean number of nodes $\mathbb{E}[V_{\alpha,T}]$**

We have

$$\mathbb{E}\left[V_{\alpha,T}\right] = \mathbb{E}\left[ \sum_i (1 - 1_{N_{ij}(T)=0, \forall j \neq i | \theta_j \leq \alpha}) \right] 1_{\theta_i \leq \alpha}$$

$$= \sum_i \left\{ 1 - \mathbb{P}(N_{ij}(T) = 0, \forall j \neq i | \theta_j \leq \alpha) \right\} 1_{\theta_i \leq \alpha}$$

Using the Palm/Slivnyak-Mecke formula and Campbell's theorem, see e.g. (Møller and Waagepetersen, 2003, Theorem 3.2) and (Kingman, 1993), we obtain

$$\mathbb{E}[V_{\alpha,T}] = \mathbb{E}\left[ \mathbb{E}\left[ V_{\alpha,T} | W_1, \ldots, W_p \right] \right]$$

$$= \mathbb{E}\left[ \sum_i \left( 1 - e^{-2T \boldsymbol{w}_i^T \sum_{j \neq i} \boldsymbol{w}_j 1_{\theta_j \leq \alpha}} \right) 1_{\theta_i \leq \alpha} \right]$$

$$= \alpha \int \mathbb{E}\left( 1 - e^{-2T \boldsymbol{w}^T \sum_j \boldsymbol{w}_j 1_{\theta_j \leq \alpha}} \right) \rho(d\boldsymbol{w})$$

$$= \alpha \int_{\mathbb{R}_+^p} \left( 1 - e^{-\alpha \psi(2Tw_1, \ldots, 2Tw_p)} \right) \rho(dw_1, \ldots, dw_p)$$

**Mean number of edges $\mathbb{E}[E_{\alpha,T}]$**

Using the extended Slivnyak-Mecke formula, see e.g. (Møller and Waagepetersen, 2003, Theorem 3.3)

$$\mathbb{E}\left[E_{\alpha,T}\right] = \mathbb{E}\left[ \mathbb{E}\left[ E_{\alpha,T} | W_1, \ldots, W_p \right] \right]$$

$$= \mathbb{E}\left[ \sum_i 1_{\theta_i < \alpha} \frac{1}{2} \sum_{j \neq i} 1_{\theta_j \leq \alpha} (1 - e^{-2T \boldsymbol{w}_i^T \boldsymbol{w}_j}) \right]$$

$$= \frac{\alpha^2}{2} \int_{\mathbb{R}_+^p} \psi(2Tw_1, \ldots, 2Tw_p) \rho(dw_1, \ldots, dw_p)$$

**Mean number of interactions $\mathbb{E}[I_{\alpha,T}]$**

We have

$$\mathbb{E}[I_{\alpha,T}] = \mathbb{E}\left[\mathbb{E}\left[I_{\alpha,T}|W_1,\ldots,W_p\right]\right]$$

$$= \mathbb{E}\left[\sum_{i\neq j}\left(\mathbb{E}\left[\int_0^T \lambda_{ij}(t)dt \mid W_1,\ldots,W_p\right]\right)\right]$$

$$= \mathbb{E}\left[\sum_{i\neq j}\left(\int_0^T \mathbb{E}\left[\mu_{ij}\frac{\delta}{\delta-\eta} - \mu_{ij}\frac{\eta}{\delta-\eta}e^{-t(\delta-\eta)} \mid W_1,\ldots,W_p\right]dt\right)\right]$$

$$= \mathbb{E}\left[\sum_{i\neq j}\boldsymbol{w}_i^T\boldsymbol{w}_j\right]\int_0^T\left[\frac{\delta}{\delta-\eta} - \frac{\eta}{\delta-\eta}e^{-t(\delta-\eta)}\right]dt$$

$$= \alpha^2\boldsymbol{\mu}_w^T\boldsymbol{\mu}_w\left(\frac{\delta}{\delta-\eta}T - \frac{\eta}{(\delta-\eta)^2}(1-e^{-T(\delta-\eta)})\right)$$

where the third line follows from (Dassios and Zhao, 2013), the last line follows from another application of the extended Slivnyak-Mecke formula for Poisson point processes and $\boldsymbol{\mu}_w = \int_{\mathbb{R}_+^p} \boldsymbol{w}\rho(dw_1,\ldots,dw_p)$.

## 3 Details of the approximate inference algorithm

Here we provide additional details on the two-stage procedure for approximate posterior inference. The code is publicly available at `https://github.com/OxCSML-BayesNP/HawkesNetOC`.

Given a set of observed interactions $\mathcal{D} = (t_k, i_k, j_k)_{k\geq 1}$ between $V$ individuals over a period of time $T$, the objective is to approximate the posterior distribution $\pi(\phi,\xi|\mathcal{D})$ where $\phi = (\eta,\delta)$ are the kernel parameters and $\xi = ((w_{ik})_{i=1,\ldots,V,k=1,\ldots,p}, (a_k, b_k)_{k=1,\ldots,p}, \alpha, \sigma, \tau)$, the parameters and hyperparameters of the compound CRM. Given data $\mathcal{D}$, let $\mathcal{Z} = (z_{ij}(T))_{1\leq i,j\leq V}$ be the adjacency matrix defined by $z_{ij}(T) = 1$ if there is at least one interaction between $i$ and $j$ in the interval $[0,T]$, and 0 otherwise.

For posterior inference, we employ an approximate procedure, which is formulated in two steps and is motivated by modular Bayesian inference (Jacob et al., 2017). It also gives another way to see the two natures of this type of temporal network data. Firstly we focus on the static graph i.e. the adjacency matrix of the pairs of interactions. Secondly, given the node pairs that have at least one interaction, we learn the rate for the appearance of those interactions assuming they appear in a reciprocal manner by mutual excitation.

We have

$$\pi(\phi,\xi|\mathcal{D}) = \pi(\phi,\xi|\mathcal{D},\mathcal{Z}) = \pi(\xi|\mathcal{D},\mathcal{Z})\pi(\phi|\xi,\mathcal{D}).$$

The idea of the two-step procedure is to

1. Approximate $\pi(\xi|\mathcal{D},\mathcal{Z})$ by $\pi(\xi|\mathcal{Z})$ and obtain a Bayesian point estimate $\widehat{\xi}$.
2. Approximate $\pi(\phi|\xi,\mathcal{D})$ by $\pi(\phi|\widehat{\xi},\mathcal{D})$.

### 3.1 Stage 1

As mentioned in Section 5 in the main article, the joint model $\pi(\mathcal{Z},\xi)$ on the binary undirected graph is equivalent to the model introduced by (Todeschini et al., 2016), and we will use their Markov chain Monte Carlo (MCMC) algorithm and the publicly available code SNetOC[1] in order to approximate the posterior $\pi(\xi|\mathcal{Z})$ and obtain a Bayesian point estimate $\widehat{\xi}$. Let $w_{*k} = W_k([0,\alpha]) - \sum_i w_{ik}$ corresponding to the overall level of affiliation to community $k$ of all the nodes with no interaction (recall that in our model, the number of nodes with no interaction may be infinite). For each

undirected pair $i, j$ such that $z_{ij}(T) = 1$, consider latent count variables $\widetilde{n}_{ijk}$ distributed from a truncated multivariate Poisson distribution, see (Todeschini et al., 2016, Equation (31)). The MCMC sampler to produce samples asymptotically distributed according to $\pi(\xi|\mathcal{Z})$ then alternates between the following steps:

1. Update $(w_{ik})_{i=1,\dots,V,k=1,\dots,p}$ using an Hamiltonian Monte Carlo (HMC) update,
2. Update $(w_{*k}, a_k, b_k)_{k=1,\dots,p}, \alpha, \sigma, \tau$ using a Metropolis-Hastings step
3. Update the latent count variables using a truncated multivariate Poisson distribution

We use the same parameter settings as in (Todeschini et al., 2016). We use $\epsilon = 10^{-3}$ as truncation level to simulate the $w_{*k}$, and set the number of leapfrog steps $L = 10$ in the HMC step. The stepsizes of both the HMC and the random walk MH are adapted during the first 50 000 iterations so as to target acceptance ratios of 0.65 and 0.23 respectively.

The minimum Bayes point estimates $(\widehat{w}_{ik})_{i=1,\dots,V,k=1,\dots,p}$ are then computed using a permutation-invariant cost function, as described in (Todeschini et al., 2016, Section 5.2). This allows to compute point estimates of the base intensity measures of the Hawkes processes, for each $i \neq j$

$$\widehat{\mu}_{ij} = \sum_{k=1}^{p} \widehat{w}_{ik} \widehat{w}_{jk}.$$

## 3.2 Stage 2

In stage 2, we use a MCMC algorithm to obtain samples approximately distributed according to

$$\pi(\phi|\widehat{\xi}, \mathcal{D}) = \pi(\phi|(\widehat{\mu}_{ij}), \mathcal{D})$$

where $\phi$ are the parameters of the Hawkes kernel. For each ordered pair $(i, j)$ such that $n_{ij} = N_{ij}(T) > 0$, let $(t_{ij}^{(1)} < t_{ij}^{(2)} < \dots < t_{ij}^{(n_{ij})})$ be the times of the observed directed interactions from $i$ to $j$. The intensity function is

$$\lambda_{ij}(t) = \widehat{\mu}_{ij} + \sum_{\ell|t_{ji}^{(\ell)}<t} \frac{\eta}{\delta} f_\delta(t - t_{ji}^{(\ell)})$$

where

$$\frac{\eta}{\delta} f_\delta(t - t_{ji}^{(\ell)}) = \eta \times e^{-\delta(t-t_{ji}^{(\ell)})}.$$

We write the kernel in this way to point out that it is equal to a density function $f_\delta$, (here exponential density), scaled by the step size $\eta/\delta$. We denote the distribution function of $f_\delta$ by $F_\delta$. Then, by Proposition 7.2.III in (Daley and Vere-Jones, 2008) we obtain the likelihood

$$L(\mathcal{D} \mid \phi, (\widehat{\mu}_{ij})) = \prod_{(i,j)|N_{ij}(T)>0} \left[ \exp(-\Lambda_{ij}(T)) \prod_{l=1}^{n_{ij}} \lambda_{ij}(t_{ij}^{(\ell)}) \right]$$

where

$$\Lambda_{ij}(t) = \int_0^t \lambda_{ij}(u)\,du = t\widehat{\mu}_{ij} + \sum_{\ell|t_{ji}^{(\ell)}<t} \frac{\eta}{\delta} F_\delta(t - t_{ji}^{(\ell)}).$$

We derive a Gibbs sampler with Metropolis Hastings steps to estimate the parameters $\eta, \delta$ conditionally on the estimates $\widehat{\mu}_{ij}$. As mentioned in Section 3 of the main article we follow (Rasmussen, 2013) for the choice of vague exponential priors $p(\eta), p(\delta)$. For proposals we use truncated Normals with variances $(\sigma_\eta^2, \sigma_\delta^2) = (1.5, 2.5)$.

The posterior is given by

$$\pi(\phi \mid \mathcal{D}, (\widehat{\mu}_{ij})) \propto \exp\left( -\sum_{(i,j)|N_{ij}(T)>0} \Lambda_{ij}(T) \right) \left[ \prod_{(i,j)|N_{ij}(T)>0} \prod_{\ell=1}^{n_{ij}} \lambda_{ij}(t_{ij}^{(\ell)}) \right] \times p(\eta)p(\delta)$$

We use an efficient way to compute the intensity at each time point $t_{ij}^{(\ell)}$ by writing it in the form

$$\lambda_{ij}(t_{ij}^{(\ell)}) = \widehat{\mu}_{ij} + \eta S_{ij}^{(\ell)}(\delta),$$

where

$$S_{ij}^{(\ell)}(\delta) = e^{-\delta t_{ij}^{(\ell)}} \sum_{k=1}^{n_{ji}} e^{\delta t_{ji}^{(k)}} 1_{t_{ji}^{(k)} < t_{ij}^{(\ell)}},$$

and then derive a recursive relationship of $S_{ij}^{(\ell)}(\delta)$ in terms of $S_{ij}^{(\ell-1)}(\delta)$. In this way, we can precompute several terms by ordering the event times and arrange them in bins defined by the event times of the opposite process.

## 4   Posterior consistency

We simulate interaction data from the Hawkes-CCRM model described in Section 3 in the main article, using parameters $p = 4, \alpha = 50, \sigma = 0.3, \tau = 1, a_k = 0.08, \phi = (0.85, 3), T = 300$. We perform the two-step inference procedure with data of increasing sample size, and check empirically that the approximate posterior $\pi(\phi | \widehat{\xi}, \mathcal{D})$ concentrates around the true value as the sample size increases. Figure 1 below shows the plots of the approximate marginal posterior distribution of $\delta$ and $\eta$. Experiments suggest that the posterior still concentrates around the true parameter value under this approximate inference scheme.

Figure 1: (Left) Approximate marginal posterior distribution of $\delta$ given $n$ interaction data. (Right) Approximate marginal posterior distribution of $\eta$ given $n$ interaction data. Posterior concentrates around the true value, with increasing sample size $n$.

## 5   Experiments

We perform experiments in which we compare our Hawkes-CCRM model to five other competing models. The key part in all cases is the conditional intensity of the point process. which we give below. In all cases we use $\{t_{ij}^{(k)}\}_{k \geq 1}$ to refer to the set of events from $i$ to $j$, i.e. the interactions for the directed pair $(i, j)$.

**Hawkes-CCRM**

For each directed pair of nodes $(i, j)$, $i \neq j$
$N_{ij}(t) \sim$ Hawkes Process$(\lambda_{ij}(t))$ where $\lambda_{ij}(t) = \sum_k w_{ik} w_{jk} + \sum_{t_{ji}^{(k)} < t} \eta e^{-\delta(t - t_{ji}^{(k)})}$.

**Hawkes-IRM (Blundell et al., 2012)**

For each directed pair of clusters $(p, q)$, $p \neq q$
$N_{pq}(t) \sim$ Hawkes Process$(\lambda_{pq}(t))$ where $\lambda_{pq}(t) = n_p n_q \gamma_{pq} + \sum_{t_{qp}^{(k)} < t} \eta e^{-\delta(t - t_{qp}^{(k)})}$.

For the details of the model see (Blundell et al., 2012).

**Poisson-IRM (as explained in (Blundell et al., 2012))**

For each directed pair of clusters $(p, q)$, $p \neq q$
$N_{pq}(t) \sim \text{Poisson}(\lambda_{pq}(t))$ where $\lambda_{pq}(t) = n_p n_q \gamma_{pq}$.

For the details of the model see (Blundell et al., 2012).

**CCRM (Todeschini et al., 2016)**

$N_{ij}(t) \sim \text{Poisson Process}(\lambda_{ij}(t))$ where $\lambda_{ij}(t) = \sum_k w_{ik} w_{jk}$.

**Hawkes**

$N_{ij}(t) \sim \text{Hawkes Process}\left(\lambda_{ij}(t)\right)$ where $\lambda_{ij}(t) = \mu + \sum_{t_{ji}^{(k)} < t} \eta e^{-\delta(t - t_{ji}^{(k)})}$.

**Poisson**

$N_{ij}(t) \sim \text{Poisson Process}(\lambda_{ij}(t))$ where $\lambda_{ij}(t) = \mu$.

## Footnotes

[1]https://github.com/misxenia/SNetOC