[Reviews · NeurIPS 2018]

Reviewer 1



Summary: This work extends the Todeschini et al. (2016) framework for sparse community structure to a temporal Hawkes process network, and thus can capture both temporal reciprocity and community structure, which are properties often studied in (temporal) social interaction data. The framework is built on using compound CRMs, and then asymptotic properties of the model are described. Lastly, empirical results for link prediction, community structure, and power law distribution are presented on a variety of temporal social network datasets. Strengths: - extends the static Todeschini et al. (2016) framework to a temporal setting, which has not been done before - Complex and flexible model for capturing sparsity, degree heterogeneity, and reciprocity, which are desirable properties for many real-world social networks with temporal structure — many existing models are either too simplistic (often ignoring temporal structure) - Develops an efficient inference algorithm for a complex model, and demonstrates on large network datasets - Well-written and good figures Weaknesses: - Can be viewed as gluing together several existing components of network models: e.g., Caron&Fox and Todeschini et al., and Blundell et al. - Inference details are sparse, may be difficult to reproduce - Experiments are simplistic and applications Quality: The paper is technically correct. The claims of sparsity, heterogeneity, and community structure follow from the inherited random measure frameworks of Todeschini and Caron&Fox, and is sparsity (asymptotic) properties are also shown in Section 4.2. The self-excitation comes from the use of the Hawkes process. The inference in the model appears to be quite scalable, as demonstrated on several large graphs. The model shows significant improvement on the link prediction task. Furthermore, it would be interesting to compare to (or even discuss more in depth) some of the more recent sparse graph models for temporal network data (e.g. Palla et al. and Ng&Silvea), which I don't believe incorporate reciprocity into the models. The paper would benefit from more of a discussion on the strengths and weaknesses of this approach for modeling real-world temporal data. Currently, it does not appear that weaknesses are addressed. The submission is clear and fairly well-written, and the figures are informative and intuitive, but in general, more details can be given as well as intuition on a dense topic. Would be useful to the reader to motivate the 2-step inference procedure a bit further. For instance, the Jacob et al. paper was motivating the modular “cutting” with robustness and model misspecification. In addition, the paper should discuss the strengths and weaknesses of this inference approach. Additionally, the inference section is a bit glossed-over in details. The current description is not the most useful for someone intending to implement the method from scratch, e.g., stating that MCMC and HMC were used is not sufficient. There is no discussion on the sensitivity of parameters and tuning that is often required. Though the inference is using an open-source package, the paper itself should include more thorough details for reproducibility purposes. Originality & related work: The main contribution of this paper is providing a new temporal network model that simultaneously captures sparsity, self-excitatory patterns, and community structure, and demonstrating inference on large graph data sets. However, all of these aspects can be viewed as a combination of well-known existing models for graphs (e.g., Caron--Fox, Todeschini et al., and Blundell et al.). The extension of the Caron--Fox and Todeschini et al. models to a temporal setting is novel, and very few models have approached this. The related work is very thorough. (The bibliography should be checked for consistency and updated, e.g., Borgs et al. is now published, inconsistencies in the NIPS conference name, etc.) Significance: I’m unsure how useful the methods in this paper are for practical network modeling, and the paper may not add much additional value beyond existing work that models self-excitatory patterns in networks (e.g., the Blundell et al. paper). Though performed on large data sets, the experiments are toy experiments that don’t carefully study the data and application, and therefore the conclusions about the data are not particularly unique or surprising. On the other hand, this is the first temporal version of the Caron--Fox framework that also includes latent community structure and also reciprocity, and could be viewed as a nice "first step" toward modeling temporal network data. --- Updated review: Overall, I think the execution appears to be fairly strong, which is why I’d given it a weak accept score. I think that it is important to have new models of the relatively-new sparse graph frameworks this paper builds on that can be applied to real network data. As this is a modeling and inference paper, the inference section needs to be able to be read on its own without referring to other work, and an honest assessment of the limitations of the model should be included. The experiments should go beyond just reporting prediction results (with error bars) and visualization of structure, and draw out the important insights of the data applications. The proofs in the appendix could also use more guiding descriptions.

Reviewer 2



I have edited my review to reflect the author feedback. Summary: This paper introduces a model for dyadic interaction data. The data consists of (t, i, j) triples where t is a timestamp and i, j are nodes/individuals in a network. The presence of a triple indicates that an interaction between i and j occurred at time t. The model does not capture direction in the base rate of interactions (i.e., \mu_{ij} = \mu_{ji}). The model assumes that each sequence of interaction events specific to an (i, j) pair is drawn from two mutually-excitatory Hawkes processes (i.e., N_{ij} and N_{ji}). The base rate of the Hawkes process is factorized under a simple CP decomposition involving a dot product between the K-dimensional vectors w_i and w_j, which each embed the nodes i and j. An interesting prior, introduced previously by Todeschini et al. (2016), is imposed over the entries of the embedding vectors; this prior encourages sparsity and power-law properties in the node degree distributions. The model is expressed using the language of point processes and completely random measures; in so doing, the paper makes clear the relationship between the proposed model and previous work. Posterior inference of the model parameters is performed using a “two-step” approximation to MCMC; this algorithm is not asymptotically guaranteed to sample from the exact posterior but can scale to larger data sets than MCMC. The paper provides some intuitive/anecdotal evidence that the scheme provides a close approximation to the exact posterior. The link prediction performance of the proposed model is compared to a series of baselines on four real-world interaction data sets; the proposed model predicts the overall number of links in the heldout portion of the data more accurately than any of the baselines (as measured by RMSE). The latent structure inferred by the proposed model is demonstrated to be interpretable; moreover the model appears to correctly capture the degree distribution and sparsity of the heldout data in the posterior predictive distribution. Detailed comments: The proposed model is elegant. Its formulation is neatly expressed and the paper cleanly situates the model in the context of previous work. Overall, the paper is well-written and a pleasure to read. While the model is elegant, I think its assumptions are too simplistic for real-world interaction data and the model itself is not novel enough to justify this. The novelty of the model stems mostly from the combination of two previously introduced components—namely, Hawkes processes and the compound CRM prior—and the approximation scheme that allows MCMC to scale. The model cannot capture any asymmetry in the base rate interaction behavior of two nodes. The base rate of events i->j is exactly the same as events j->i. Many real-world data sets that feature directed interactions also exhibit this kind of asymmetry. In order to capture that kind of asymmetry the model could make the dot product (in equation 9) into a bilinear form involving a middle matrix e.g., as in Zhou (2015) [1] (which is a paper that should be cited alongside the other papers on edge-exchangeable models). The model makes the assumption that the event sequences are mutually-excitatory which does capture "reciprocity"; however the fact that the model is able to capture reciprocity, sparsity, degree heterogeneity, and community structure is not surprising; these properties have already been shown to emerge independently from the two pieces the proposed model combines. Question: —The results reported in Table 2 are reported to be “the root mean square error between the predicted and true number of interactions in the test set”. Does this mean the RMSE between the pointwise counts N_{ij} for each pair (i, j) that was heldout? Or does this mean that *ALL* events in the test set were summed and compared to the prediction of the sum of all heldout events. The latter would be strange (but is most consistent with the wording). Please clarify. [1] Zhou, Mingyuan. "Infinite edge partition models for overlapping community detection and link prediction." Artificial Intelligence and Statistics. 2015.

Reviewer 3



The paper considers Hawkes processes with intensities constructed from generalized gamma process base intensity functions. Many desired properties for dynamic network models (particularly applied to social networks) are shown to be captured. A strong paper. It runs along a current thread of work on such models that are somehow inspired by, or connected to, the edge-exchangeable models derived from point processes. So it may not be terribly impactful in that sense, but it is still a novel, elegant piece of work, building upon several sophisticated tool sets (Hawkes processes, CRMs in the generalized gamma class, edge-exchangeable models, dynamic network constructions), it is excellently written and the experiments are broad and convincing. The paper has plenty of color and hits all the right spots, making a satisfying NIPS read. I don't find the inference section very reproducible. Perhaps at the very least you should provide more details in the supplement. You're using a lot of pieces from different pieces of work, so it seems that it would be very difficult for something to get all these elements working well on their own. If accepted, I'm tempted to check back in on the final submission and kick up a fuss if it isn't satisfied... Other comments: p. 2, line 79, "...in that specific case...": What specific case? Don't know what you're talking about here. p. 6, equation after line 174: I don't get the first equality. How can you say these two densities are equal? Surely you're missing a change to the conditioning set or are hiding an integration. p. 6, inference: Again, I REALLY don't think I could implement inference procedure from just reading this section... and I'd consider myself pretty well-versed on these... p. 7, "link prediction" experiments: So am I understanding this correctly... You're only predicting the total number of interactions in the test set? So just one number? And table Table 1 just reports RMSE on this ONE number? I would have been a bit more satisfied if you at the very least made multiple train/test splits, perhaps over sliding windows or something. I find the experiments strong in general but this specific point is a bit of let down to me. p. 7, line 245, "... can uncover the latent community structure...": Well not THE latent community structure; perhaps A community structure. In general this is non-identifiable and not all samples would be intuitively/qualitatively interpretable, I would imagine. Updated review: After viewing the author responses, I maintain my strong accept recommendation. I do still believe this would be in the top 50% of accepted NIPS papers. The only element in the author response relevant to my comments were regarding the inference section; I trust the authors' promises to include more detail in the supplementary material. In their rebuttal, the authors state a large amount of the code from Todescini et al. was used... I think this should be emphasized in the paper... A major goal is to present reproducible work, and if a reader knows much of the heavy lifting will be done by an existing code base, I think that would highly encourage someone to pick up your work.